# Boron-chalcogen heterocycles and linear tetraboranes from a cyclic tetra(amino)tetraborane

Eva Beck[1,2], Diana Bröllos [1,2], Ivo Krummenacher [1,2], Thomas Kupfer[1,2], Maximilian Dietz [1,2], Tim Wellnitz[1,2], Cornelius Mihm[1,2] & Holger Braunschweig [1,2] ✉

While small carbocyclic rings have long been recognized as pivotal building blocks in chemistry, their all-boron counterparts have remained largely unexplored. In this work, we present a detailed account of the functionalization reactivity of our cyclic tetraborane $B_4(NCy_2)_4$ (Cy = cyclohexyl) encompassing both ring-expansion and ring-opening reactions. Specifically, diphenyl dichalcogenides effect ring expansion to five-membered $B_4E$ rings (E = S, Se, Te), while halogenating agents induce ring opening to generate linear tetraboranes with halide end groups. These transformations reveal reactivity patterns reminiscent of strained organic ring systems, thus highlighting the cyclic tetraborane's potential as a versatile precursor for synthesizing intricate boron-rich architectures.

Small ring systems play a crucial role in chemical synthesis, serving as fundamental scaffolds in molecular architectures and as key components in pharmacologically active compounds[1–11]. Their unique structural characteristics, and their inherent ring strain in particular, allow them to function as versatile synthetic intermediates, enabling diverse chemical transformations, including ring expansions, desymmetrizations, and ring-opening processes critical for constructing complex molecular structures[1–7]. While these traits are well-established for three- and four-membered rings based on carbon, boron-containing analogs have largely remained unexplored, only recently gaining more prominence in scientific research[12–17].

The synthesis of small boron rings presents intricate challenges stemming from both structural and electronic characteristics. With only three covalent bonds, boron leaves an empty p orbital that creates an incomplete octet, thus rendering the atom inherently electron-deficient. This electronic configuration drives boron's propensity for multicenter bonding, which fundamentally destabilizes conventional ring structures abundantly found for cyclic hydrocarbons[18,19]. A cyclic triborane of the form $B_3R_3$ with σ-bonds between the boron atoms, i.e., a structural analog of cyclopropane, still remains an elusive target. Yamamoto's

reported triaminotriborane(3) exemplifies these challenges[20]: despite employing electron-donating amino substituents to alleviate electron-deficiency and promote classical bonding, the compound adopts a bent chain structure in the solid state with delocalized 3c-2e π-bonds rather than a closed three-membered $B_3$ ring. Density functional theory (DFT) analysis suggests that the bent chain configuration primarily stems from the steric hindrance of the bulky TMP ligands (TMP = 2,2,6,6-tetramethylpiperidino), as the sterically less encumbered dimethylamino derivative is calculated to prefer a three-membered ring structure. Similar complexities emerged in attempts to create a neutral tricycloborane(3) through two-electron oxidation of the 2π-aromatic triboracyclopropenyl dianion (**A**, Fig. 1), in which case the $B_3$ ring, stabilized by dicyclohexylamino substituents, proved unstable and engaged in ring-opening reactions[21]. While a neutral cyclotriborane(3) remains uncharacterized, two closely related tetraborane(4) derivatives with $B_4$ rings have been successfully isolated: a diisopropylamino-substituted cyclotetraborane from the group of Siebert[22], and a dicyclohexylamino derivative from our own research (**B**, Fig. 1)[23,24]. The former was generated by reductive homocoupling of a diborane(4) precursor, while the

[1]Institut für Anorganische Chemie, Julius-Maximilians-Universität Würzburg, Am Hubland, Würzburg, Germany. [2]Institute for Sustainable Chemistry & Catalysis with Boron, Julius-Maximilians-Universität Würzburg, Am Hubland, Würzburg, Germany. ✉e-mail: h.braunschweig@uni-wuerzburg.de

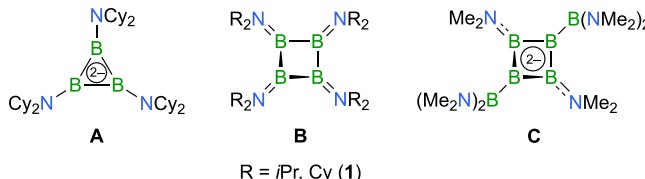

**Fig. 1 | Classical all-boron ring systems.** Examples of small homocyclic boron rings featuring classical bonding patterns known in the literature: the triboracyclopropenyl dianion **A**; cyclotetraboranes **B**; the cyclotetraborane dianion **C**.

**Fig. 2 | Inserting chalcogen atoms – ring expansion reactivity of 1.** Reactions of $B(NCy_2)_4$ (**1**) with diphenyl dichalcogenides selectively produce $B_4E$ (E = S, Se, Te) ring systems **2S**, **2Se**, and **2Te** via chalcogen atom insertion and ring expansion.

latter emerged from the reduction of a dihaloborane that led to the catenation of four boron atoms. Both boranes (**B**) adopt a folded structure similar to that of the hydrocarbon cyclobutane. Derivatives with different ring substituents, such as TMP groups or alkyl and halogen substituents, illustrate the delicate balance governing ring stabilization and the inherent constraints on functional group diversity in boron rings[22,25–27]. In each of these derivatives, the boron atoms reorganize into a tetrahedral structure characterized by electron-deficient multicenter bonding. With mixed boryl and amino substituents, yet another structural motif emerges in a neutral tetraborane(4)[28]. Derived from the dianionic 2π-aromatic compound **C** (Fig. 1)[29,30], this configuration adopts a non-classical planar rhomboid structure with delocalized σ- and π-bonding that preserves its aromatic character[28].

Considering the intricate effects of substituents on structural preferences, tetraboranes(4) with an electron-precise four-membered $B_4$ ring are rare, leaving their chemistry largely unexplored. In a recent communication, we detailed the fundamental redox chemistry of compound **1**, which highlighted its unique ability to undergo both reduction and oxidation while preserving its distinctive folded ring structure[23]. Here we report structurally confirmed ring-expansion and ring-opening transformations of tetraborane **1**, thus elucidating two nuances of small boron cycles' basic chemistry in great detail. Chalcogen atom insertion generates five-membered $B_4E$ rings (E = S, Se, Te), while halogenating agents induce ring scission to form linear (pseudo)halide-functionalized tetraboranes.

## Results and discussion

### Ring-expansion reactivity

Given its small ring structure, we anticipated that tetraborane **1** would readily engage in a variety of ring-expansion and ring-opening

reactions. We first examined its reactivity with elemental chalcogens (S and Se), and found that compound **1** underwent largely unselective reactions after their initiation at elevated temperatures. In addition to the expected chalcogen atom insertions producing five-membered $B_4E$ rings (E = S and Se), several other products were identified through ${}^{11}B$ NMR spectroscopy. These included 1,3-dichalcogeno-2,4-diboretanes with the general formula $[Cy_2NBE]_2$ (E = S and Se), as will be discussed later in this section.

In an effort to achieve a more selective insertion process, we explored alternative chalcogen sources. Here, we found that diphenyl disulfide ($Ph_2S_2$), diphenyl diselenide ($Ph_2Se_2$), and diphenyl ditelluride ($Ph_2Te_2$) generated the corresponding ring-enlarged products, i.e. 1-thia-, 1-selena-, and 1-tellura-2,3,4,5-tetraborolanes **2S**, **2Se**, and **2Te**, with high selectivity in good to excellent isolated yields (Fig. 2). Sulfur insertion into **1** using diphenyl disulfide was most effectively accomplished via UV irradiation (210–600 nm) of a benzene solution at room temperature for 2 d. The resulting five-membered product **2S** was collected as a colorless solid in a yield of 81%. **2S** displays two distinct ${}^{11}B$ NMR signals at $\delta = 55.5, 45.9$ ppm. Crystals suitable for X-ray diffraction analysis were readily obtained by cooling a saturated hexane solution to −30 °C. In the solid state, the $B_4S$ ring adopts a twisted conformation, in which only the B – S – B atoms lie in one plane, while the other two boron atoms are positioned above and below this plane, respectively (Fig. 3). The bond angle at sulfur is 95.56(6)°. The boron-sulfur bond distances of 1.871(1) and 1.876(1) Å are consistent with single bonds, as are the B – B bond lengths, which show values between 1.702(2) and 1.727(2) Å. The boron atom that deviates most from ideal trigonal planarity (B3, Fig. 3), with a sum of bond angles of 353.8°, exhibits a slightly longer boron-nitrogen bond (1.430(8) Å) than those of the other three B atoms (1.408(2), 1.405(2), 1.40(1) Å). While examples of five-membered rings containing only boron and sulfur atoms are known[31–35], a $B_4S$ ring featuring a chain of four boron atoms, as seen in **2S**, has remained absent in the literature thus far.

We observed that half an equivalent of diphenyl disulfide was sufficient to fully convert **1** into **2S**, which suggests that both of its sulfur atoms are incorporated into the product. The elimination of one sulfur atom would formally produce diphenyl sulfide ($Ph_2S$), which might subsequently act as a secondary chalcogen source. Indeed, when **1** was reacted with diphenyl sulfide under UV irradiation, **2S** was isolated in 74% yield. Both sulfide and disulfide reagents can generate phenylthiyl radicals under these conditions[36,37]; however, their involvement in product formation remains uncertain and warrants further investigation. It is important to highlight that no additional insertion of chalcogen atoms was observed, even when an excess of the diphenyl dichalcogenide reagent was used.

The heavier chalcogen analogs of **2S** are readily available by reaction of **1** with diphenyl diselenide and diphenyl ditelluride under thermal or photolytic conditions, respectively (Fig. 2). Similar to the sulfur derivative, **2Te** is best prepared by UV irradiation of a mixture of **1** and $Ph_2Te_2$ in a benzene solution for 1 d. By contrast, the selenium derivative **2Se** is most efficiently generated by heating **1** with $Ph_2Se_2$ at 80 °C for 2 d. Both compounds were isolated as colorless solids, with ${}^{11}B$ NMR chemical shifts of $\delta = 55.4, 46.8$ ppm (**2Se**), and $\delta = 58.7, 44.5$ ppm (**2Te**), placing them in a region similar to that of **2S**. The chalcogen atoms are observed with chemical shifts of $\delta({}^{77}Se) = 254$ ppm, and $\delta({}^{125}Te) = 15.6$ ppm. In the solid state, the $B_4E$ rings of both **2Se** and **2Te** adopt the same twisted conformation as observed for **2S**, with chalcogen bond angles near 90° (**2Se** 92.6(4)°; **2Te** 88.43(6)°; Fig. 3). The boron-boron and boron-nitrogen distances are comparable to those in **2S** but are more uniform. Alongside **2S**, these boron-chalcogen heterocycles featuring four boron atoms and a single chalcogen atom represent an intriguing class of inorganic ring compounds that was previously unknown. The four boron atoms in the ring's backbone are interconnected by classical B – B single bonds, a structural feature presumably attributed to efficient electron donation

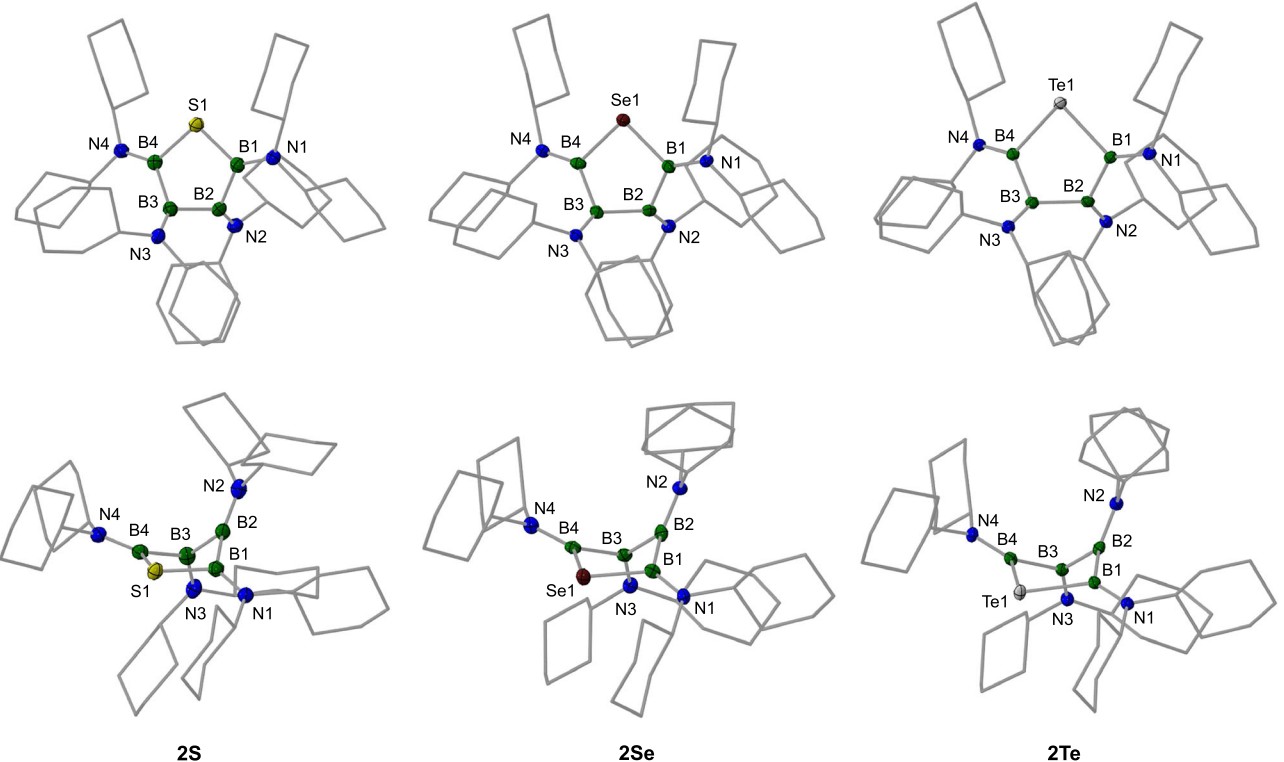

**Fig. 3 | Structures of B₄E heterocycles (E = S, Se, Te) in the solid state.** Molecular structures of **2S**, **2Se**, and **2Te** are presented from both top (top) and side views (bottom). Displacement ellipsoids shown at the 50% probability level; hydrogen atoms and ellipsoids of the cyclohexyl groups are omitted for clarity.

from the amino substituents that inhibits the boron atoms from participating in nonclassical bonding[22–24].

As mentioned at the beginning of this section, the reactions of **1** with elemental chalcogens lack selectivity, yielding a variety of products (Fig. 4). For instance, treating **1** with sulfur in a toluene solution at 85 °C for one week resulted in the appearance of several new resonances in the $^{11}$B NMR spectrum. In addition to some unidentified species ($\delta$($^{11}$B) = 43.3, 40.5, 30.8, 28.5 ppm), the boron-sulfur heterocycle **2S** and an additional compound with a shift of $\delta$ = 37.5 ppm were identified. Mass spectrometry and X-ray crystallography confirmed this latter compound as [Cy₂NBS]₂ (**3S**), a 1,3-dithia-2,4-diboretane with structural precedent in the literature (see Supplementary Information for details)[38]. It could be isolated from the mixture of products in 15% yield. Other products eluded accurate quantification, as only very low amounts of crystalline material were obtained. The same dithiadiboretane product was identified in reactions of **1** with trimethylphosphine sulfide (Me₃PS) as the chalcogen source. Similar results were found with elemental selenium, for which a range of $^{11}$B NMR signals emerged from the reaction mixture. Only the five-membered ring product (**2Se**) and 1,3-diselena-2,4-diboretane **3Se** ($\delta$($^{11}$B) = 35.5 ppm) were identified unambiguously, although the latter could not be characterized by X-ray crystallography. Reaction of **1** with trimethylphosphine selenide (Me₃PSe) proved to be unselective as well. For tellurium, ring expansion proved unsuccessful entirely; **1** was inert toward elemental tellurium, and its reaction with tri-*n*-butylphosphine telluride (*n*Bu₃PTe) produced inseparable product mixtures.

In our pursuit of the lightest chalcogen derivative, i.e., **2 O**, we tried several approaches to incorporate an oxygen atom into **1** through ring expansion. Surprisingly, tetraborane **1** was found to be remarkably stable under ambient conditions and does not readily react with molecular oxygen or moisture. Both in the solid state and in solution, **1** remains unaffected for several days without any signs of decomposition. In an effort to induce a reaction with oxygen, we

allowed a solution of **1** in benzene to react for 4 d at room temperature in a 1 bar oxygen atmosphere. Monitoring the reaction by NMR spectroscopy revealed the emergence of several $^{11}$B NMR resonances, suggesting an unselective reaction. In the reaction mixture, we identified the signal for the boroxine [Cy₂NBO]₃ ($\delta$($^{11}$B) = 20.9 ppm), which was further characterized by X-ray crystallography, as its structural data was previously lacking in the literature[39]. To promote a more selective and controlled oxidation reaction of **1**, we also used other reactive oxidizing agents such as trimethylamine *N*-oxide (Me₃NO), nitrous oxide (N₂O), and *meta*-chloroperbenzoic acid (*m*CPBA). While an unselective oxidation reaction was induced with *m*CPBA at room temperature, Me₃NO and N₂O showed no conversion under the same conditions. However, heating these solutions at 80 °C led to the formation of the corresponding boroxine alongside other unidentified products.

Chalcogen atom insertions into B – B single bonds, while rare, have been documented for selected diborane(4) systems[40–42]. Himmel et al. achieved sulfur insertion using both elemental sulfur and diphenyl disulfides[40], while Ghosh and coworkers demonstrated both sulfur and selenium insertions[41]. Our group reported oxygen atom insertion into a dialkynyl-substituted diborane(4) using Me₃NO[42]. Similar to the reactivity of cyclic tetraborane **1** towards elemental chalcogens described here, we have previously also observed rather unselective chalcogen insertions into an amino-substituted triborane(5), yielding a complex mixture of insertion products that proved difficult to isolate[43].

## Redox chemistry of the B₄E rings

As a new class of boron-chalcogen heterocycles, **2E** (where E = S, Se, and Te), we explored their fundamental chemical and redox properties. Our findings reveal that these compounds exhibit remarkable thermal stability and resistance to chemical attack. The heterocycles **2E** are stable under an oxygen atmosphere, even upon heating at 80 °C for several days, and towards other oxygenation reagents such as

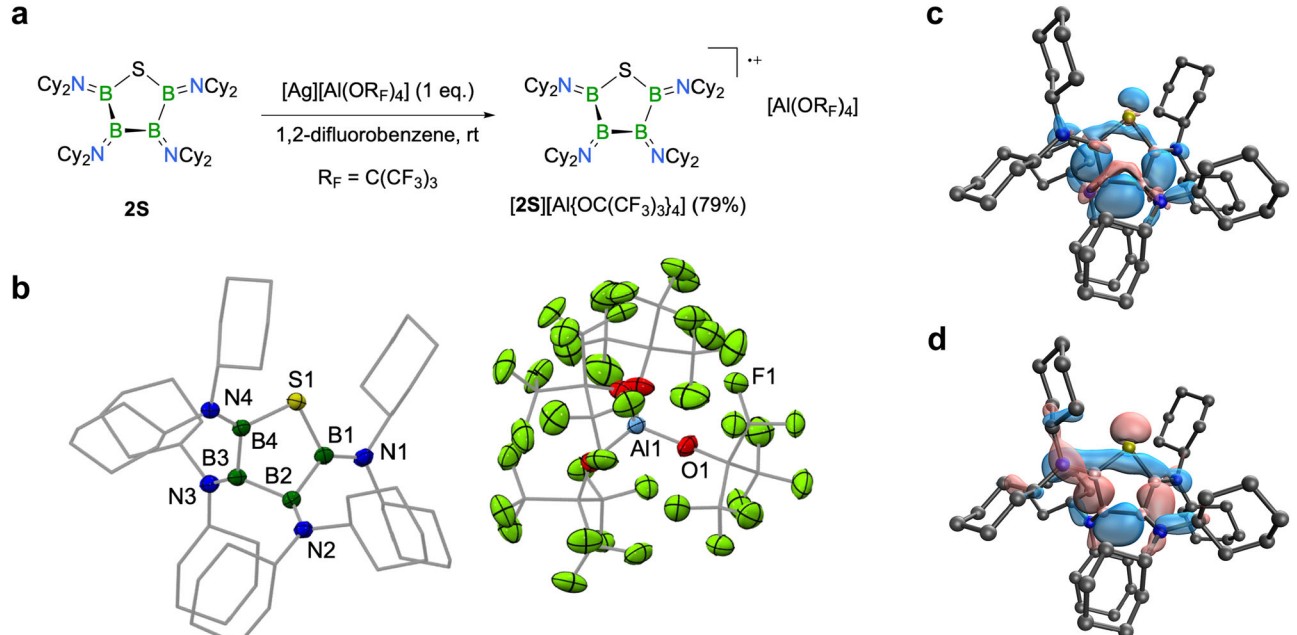

**Fig. 4 | Lack of selectivity for elemental chalcogens as reagents.** Attempts aiming at ring expansion of tetraborane **1** via chalcogen atom insertion into B−B bonds using elemental chalcogens as reagents are unselective.

**Fig. 5 | Synthesis, structure, and electronic nature of the radical cation [2S]⁺.** **a** One-electron oxidation of **2S** to afford [**2S**][Al{OC(CF₃)₃}₄]. **b** Molecular structure of [**2S**][Al{OC(CF₃)₃}₄] in the solid state. Displacement ellipsoids shown at the 50% probability level; hydrogen atoms and ellipsoids of the cyclohexyl groups are omitted for clarity. **c** Spin density plot of [**2S**]⁺ (isosurface plot at 0.004 a.u.), and **d** SOMO of [**2S**]⁺ (isosurface plot at 0.04 a.u.), illustrating the importance of σ-electron delocalization effects (SMD-ωB97xd/def2-TZVP).

Me₃NO. They also demonstrate high stability toward strong Brønsted acids ([H(OEt₂)₂][BArᶠ₄], Arᶠ = 3,5-(CF₃)₂C₆H₃)[44,45], as well as reduction and the coordination of Lewis acids (e.g. AlCl₃, GaCl₃, AlMe₃). Their robust nature is further exemplified by their inertness towards phosphines (PMe₃, PPh₃), which failed to abstract the chalcogen atom to reform **1**. However, our studies showed that these heterocycles are susceptible to oxidative B−B bond cleavage, yielding ring-opened, difunctionalized borane derivatives.

Electrochemical studies of **2S** in THF showed a reversible oxidation process ($E_{1/2}$ = 0.23 V vs. Fc⁺/⁰) in its cyclic voltammogram, indicating the formation of a stable radical cation species. Hence, treatment of a 1,2-difluorobenzene solution of **2S** with one equivalent of [Ag][Al{OC(CF₃)₃}₄][46] produced a dark green solution, from which the radical cation [**2S**][Al{OC(CF₃)₃}₄] was isolated in 79% yield (Fig. 5a). The EPR spectrum of this species in a mixture of toluene and 1,2-difluorobenzene features a broad, poorly resolved signal at a *g* factor of ca. 2.0022. Quantum chemical calculations at the SMD-ωB97xd/def2-TZVP//ωB97xd/def2SVPP level of theory[47] suggest that the spin density is delocalized predominantly along the σ-framework of the B₄ entity within the ring; a spin density plot of [**2S**]⁺ is depicted in Fig. 3c. This effect is also illustrated nicely by the topology of the singly occupied molecular orbital (SOMO) of [**2S**]⁺, which shows significant contributions from all three B−B σ-bonds (Fig. 5d). We note that similar results were recently obtained with the radical cation of **1**, [**1**][Al{OC(CF₃)₃}₄], in parallel studies[23]. X-ray diffraction analysis

confirmed the structure to be the radical cation of **2**, with the five-membered ring remaining intact in the +1 oxidation state (Fig. 5b). This single-electron oxidation of **2S** induces notable structural changes: B−N bonds (1.366(4)−1.388(3) Å) contract, B−B (1.737(4)−1.785(4) Å) and B−S bonds (1.843(2)−1.846(3) Å) elongate, and the B−S−B angle widens slightly (97.1(1)°). We also note that all our efforts to generate a dication of **2S** by applying two or more equivalents of [Ag][Al{OC(CF₃)₃}₄] failed thus far, and only [**2S**][Al{OC(CF₃)₃}₄] could be detected.

No radical species was detected for the reaction of **2S** with silver trifluoromethanesulfonate (AgOTf). Instead, we observed ring opening caused by nucleophilic attack of the triflate anion to the boron ring atoms. When treated with two equivalents of AgOTf in a benzene solution at 50 °C, we were able to isolate a colorless solid product with a yield of 33% (Fig. 6). Characterization of single crystals of the product, **4S**, by X-ray crystallography indicated the cleavage of the B−B bond at the ring site opposite the sulfur center, and the formation of a ring-opened product with triflate groups at each end. In this linear chain form, the bond angle at sulfur is widened to nearly 119.1(1)°, while the B−S bond distances (1.876(3), 1.860(3) Å) are unaltered from those in the B₄S ring of **2S**. The boron-boron bond distances in **4S** (1.718(4), 1.707(4) Å) also fall within the range observed in **2S**. The B−N bonds involving the triflate-bearing terminal boron atoms are slightly longer than the internal ones, which remain similar to those in **2S**. The B−O bond length of 1.466(3) Å aligns well with those observed in other

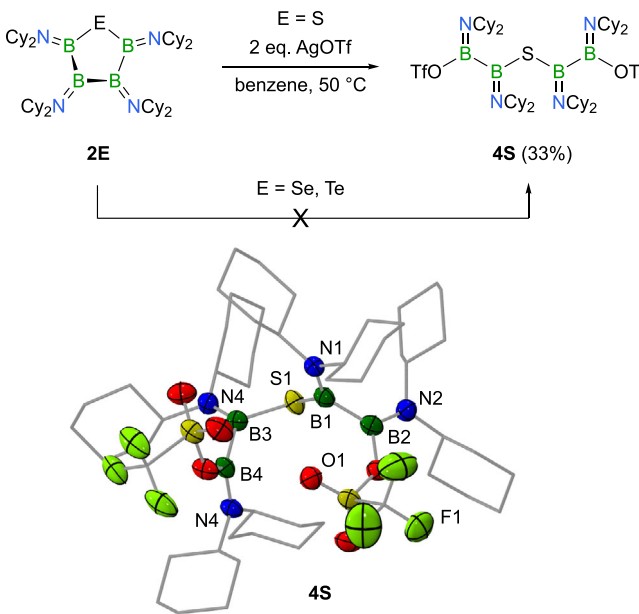

**Fig. 6 | Ring-opening reactivity of 2E (E = S, Se, Te).** Synthesis of linear **4S** via selective oxidative ring-opening of **2S** by AgOTf, along with its molecular structure (displacement ellipsoids at 50% probability level; hydrogen atoms and ellipsoids of the cyclohexyl groups omitted for clarity). For E = Se, Te, experiments targeting similar ring-opening events were either unselective or failed at all.

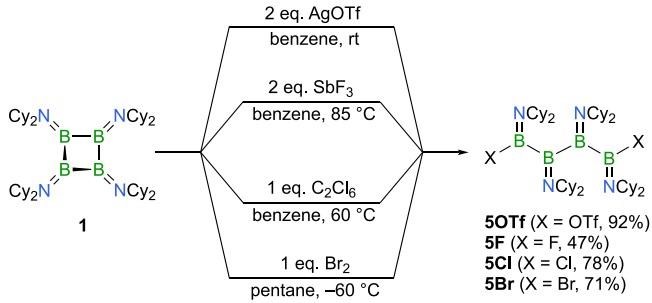

**Fig. 7 | Ring-opening reactivity of 1.** Selective ring-opening processes of **1** – formation of linear tetraboranes **5F**, **5Cl**, and **5Br** by reaction with halogenating agents.

triflate-containing boranes featuring three-coordinate, planar boron centers[48,49]. Variable-temperature NMR studies of **4S** in toluene solution are reminiscent of dynamic behavior, thus suggesting the presence of multiple conformations. While two distinct $^{11}$B NMR resonances are observed from room temperature down to 193 K, the $^{19}$F NMR spectra show increasing complexity with the emergence of additional signals, indicating the presence of distinct conformational species in slow exchange. Heating the toluene solution initially decreased the linewidths of the $^{11}$B NMR signals, but ultimately led to rapid product decomposition. While numerous cyclic compounds containing a B − S − B moiety are documented in the literature, acyclic analogs featuring a structure such as **4S**, remain rare. Only one example, a diboryl sulfide reported by Nöth as a side product from chloroborane reduction, has been previously characterized[50]. A related structure featuring a six-membered $B_4S_2$ ring with B − B − S − B − B connectivity was also reported by the same group[31]. Interestingly, analogous oxidation reactions of the higher homologues **2Se** and **2Te** with AgOTf (2 eq.) did not yield any reaction under the same conditions (Fig. 6), which is particularly noteworthy given that cyclic voltammetry indicated similar first oxidation potentials (**2Se** $E_{1/2}$ = 0.20 V, **2Te** $E_{1/2}$ = 0.16 V; vs. Fc$^{+/0}$). However, **2Se** could be forced to react with

AgOTf by applying higher temperatures (80 °C) and significantly longer reaction times (6 days), but proved highly unselective and produced no tractable materials. **2Te**, by contrast, showed no reactivity toward AgOTf at all, even at 100 °C. The reason for these differences is unknown to us so far.

## Ring-opening reactivity of 1

The successful B−B bond cleavage of **2S** upon pseudohalogenation to form **4S** led us to explore whether cyclic tetraborane **1** can also be used to generate 1,4-difunctionalized tetraboranes. Initial investigations focused on AgOTf, followed by studies with conventional halogenating agents such as bromine ($Br_2$), hexachloroethane ($C_2Cl_6$), and antimony trifluoride ($SbF_3$). It should be noted that, similar to **2S**, cyclic tetraborane **1** undergoes one-electron oxidation to form a radical cation, which we previously isolated using the weakly coordinating polyfluoro-alkoxyaluminate $[Al\{OC(CF_3)_3\}_4]^-$ counterion[23]. While reactions with other silver salts of different anions were generally unselective, we discovered that treatment of **1** with two equivalents of AgOTf in benzene yields the ring-opened 1,4-bistriflate derivative **5OTf** as a colorless solid in 92% yield (Fig. 7). Although X-ray diffraction analysis confirmed the connectivity of **5OTf**, extensive disorder in the crystal structure precluded detailed structural analysis. Solution NMR studies suggested conformational dynamics similar to those observed for **4S**, as evidenced by four distinct $^{19}$F NMR signals for the $CF_3$ groups, which coalesce into three signals at lower temperatures (see Supplementary Information for details). Its $^{11}$B NMR spectrum features two broad signals at $\delta$ = 55.5 and 36.2 ppm for the inner and outer boron atoms, respectively.

Using $SbF_3$, $C_2Cl_6$, and $Br_2$ as halogenation agents, we successfully transferred this reactivity to the respective halide-substituted tetraboranes **5F**, **5Cl**, and **5Br**, which were isolated in good yields ranging from 47 to 78% (Fig. 7). Again, each of these compounds show two distinct $^{11}$B NMR signals for the four boron atoms; the outer two boron atoms at lower chemical shifts ($\delta$($^{11}$B) = 37.9–43.4 ppm), the inner two boron atoms in a narrow region between $\delta$ = 56.9 and 57.5 ppm. The presence of fluoride served as a reliable spectroscopic probe for verifying conformational dynamics in solution for **5F** similar to **4S** and **5OTf**. Thus, **5F** shows five distinct $^{19}$F NMR signals in a range between $\delta$ = −76.18 and −88.53 ppm. While these dynamics are directly detectable only in fluorine-containing species, it seems reasonable to conclude that analogous conformational behavior exists for the other halide derivatives as well. To gain better insights into these dynamics, we exemplarily studied the thermodynamics of four hypothetical conformers of **5F** by theoretical methods, all of them derived from its "gauche"-type X-ray structure (see below). To this end, we optimized **5 F** (**5F**$^{conf1}$), and generated conformers by 180° rotations around the B2 − B3 bond (**5F**$^{conf2}$), the B3 − B4 bond (**5F**$^{conf3}$), or a combination of both operations (**5F**$^{conf4}$; see Supplementary Information for details). As expected, the conformers are close in energy, with the experimentally verified "gauche"-type arrangement **5F**$^{conf1}$ being thermodynamically slightly favored. The other conformers display only marginally higher energy levels: **5F**$^{conf4}$ $\Delta E$ = +0.3 kcal/mol; **5F**$^{conf3}$ $\Delta E$ = +2.1 kcal/mol; **5F**$^{conf2}$ $\Delta E$ = +5.4 kcal/mol. While we have not investigated the barriers between conformers, their comparable thermodynamic stability supports our hypothesis of conformational dynamics. We also note that the calculated $^{19}$F NMR chemical shifts of the conformers are found in a similar range as the experimental signals (see Supplementary Information for details). The molecular structures of all derivatives confirm their open-chain structure (Fig. 8). All species adopt a "gauche" conformation, characterized by dihedral angles of around 60° between the BX(NCy$_2$) groups (X = F, Cl, Br). Specifically, the angles measured are 46.9(1)° for the fluorine derivative **5 F**, 74.9(3)° and 72.5(3)° for the chlorine derivative **5Cl**, and 63.6(2)° for the bromine derivative **5Br**. The nearly perpendicular arrangement of the trigonal planes of the four boron atoms results in either a syn or

anti relationship between the halogen substituents. While the fluoride and chloride derivatives adopt an anti configuration, the bromides in **5Br** show a syn arrangement. This orientation induces statistically significant differences in the B − N and B − B bonds. In **5Br**, the B − N bonds on the terminal boron atoms are marginally shorter than the central B − N bonds, while the outer B − B bonds are measurably shorter than the central B − B bond, reflecting the influence of the electronegative halide substituents.

Recently, we generated related 1,4-difunctionalized tetraboranes by halogenation of linear hexakis(dimethylamino)tetraborane(6), $B_4(NMe_2)_6$[51]. Despite the overall similarity in spectroscopic profiles, a notable difference is evident in the NMR data of its OTf derivative, which displays a single $^{19}F$ NMR resonance at $\delta = -77.43$ ppm. This observation suggests a higher degree of symmetry in solution, presumably due to a dynamic equilibrium of conformers. The rapid interconversion of these conformers is likely facilitated by the low steric hindrance associated with the $NMe_2$ substituents, minimizing any steric interactions hindering free rotation around the boron-boron bonds. It is also worth noting that **5 F** represents the first example of a fluoride-substituted derivative.

By contrast to the reactivity described above, 1,2-dibromo-1,2-bis(dicyclohexyl-amino)diborane(4) **6** is formed instead of the linear tetraborane **5Br**, when **1** is reacted with an excess of elemental bromine $Br_2$ (Fig. 9). After work-up, **6** is isolated in a moderate yield of 56%, easily identified by a $^{11}B$ NMR chemical shift of $\delta = 37.8$ ppm, which is nearly identical to that of the related $NMe_2$-substitued diborane(4) ($\delta(^{11}B) = 37.7$ ppm)[52]. In addition to multinuclear NMR spectroscopy, **6** was characterized by solid-state X-ray diffraction and mass spectrometry. Similar to the dimethylamino derivative, the angles between the planes of the boron atoms are nearly perpendicular, forming an angle of 88.6°. Also, bond lengths for B − N (1.382(2), 1.384(2) Å), B − B (1.691(3) Å), and B − Br (1.978(2), 1.982(2) Å) are comparable. No further oxidation of **6** to the corresponding amino(dibromo)borane $(Cy_2N)BBr_2$ was observed regardless the amount of $Br_2$ used in the bromination of **1**. Also, no related overchlorination reactivity was observed during the reaction of **1** with hexachloroethane, and we did not detect any evidence for the generation of the corresponding 1,2-dichlorodiborane(4). However, we note that selective oxidation of **1** to **5Cl** was also achieved when **1** was reacted with one equivalent of $AlCl_3$ and $GaCl_3$. With its confirmed ring-opening reactivity towards halogenating agents, cyclotetraborane **1** thus exhibits typical behavior associated with small ring systems. This reactivity makes **1** a valuable building block for the synthesis of 1,4-difunctionalized tetraboranes, which are otherwise challenging to produce, thereby opening new avenues for developing complex boron-chain-containing compounds.

We have demonstrated that tetrakis(dicyclohexylamino)cyclotetraborane, $B_4(NCy_2)_4$, readily participates in ring-expansion reactions with diphenyl dichalcogenides to form non-planar five-membered $B_4E$ heterocycles (E = S, Se, Te). The sulfur derivative shows interesting reactivity patterns, including oxidation to a radical cation and ring opening with silver triflate, which generates a rare diboryl sulfide. Similar reactivitiy was observed with the cyclic tetraborane precursor, which reacted with various halogenating reagents to produce synthetically valuable, linear tetraboranes functionalized at the termini with triflate, fluoride, chloride, and bromide. The steric hindrance of the dicyclohexyl groups restricts rotation around the boron-boron bonds, leading to the formation of various conformers in solution, as suggested by $^{19}F$ NMR spectroscopy and DFT calculations. Further investigations into the reactivity of the cyclic tetraborane are currently underway in our laboratory, with a particular focus on enhancing our understanding of boron-rich compounds that exhibit classical bonding between boron atoms.

## Methods
### General
All manipulations were performed either under an atmosphere of dry argon or in vacuo using standard Schlenk line or glovebox techniques. Deuterated solvents were dried over molecular sieves and degassed by three freeze-pump-thaw cycles prior to use. All other solvents were distilled and degassed from appropriate drying agents. Both deuterated and non-deuterated solvents were stored under argon over activated 4 Å molecular sieves. Liquid-phase NMR spectra were acquired on a Bruker Avance 400 ($^1H$: 400.1 MHz, $^{11}B$: 128.5 MHz, $^{13}C$: 100.7 MHz), Bruker Avance 500 ($^{11}B$: 160.5 MHz, $^{13}C$: 125.7 MHz), or Bruker Avance 600 ($^1H$: 600.1 MHz, $^{11}B$: 192.6 MHz, $^{13}C$: 150.9 MHz, $^{19}F$: 564.7 MHz, $^{77}Se$: 114.5 MHz, $^{125}Te$: 189.3 MHz) spectrometers. Chemical shifts ($\delta$) are reported in ppm and internally referenced to the carbon nuclei ($^{13}C\{^1H\}$) or residual protons ($^1H$) of the solvent. Heteronuclear NMR spectra are referenced to external standards ($^{11}B$: $BF_3 \bullet OEt_2$, $^{19}F$: $CCl_3F$, $^{77}Se$: $Me_2Se$, $^{125}Te$: $Me_2Te$). Resonances are given as singlet (s), multiplet (m) or broad (br). High-resolution mass spectrometry (HRMS) data were obtained from a Thermo Scientific Exactive Plus spectrometer. EPR measurements at X-band (9.86 GHz) were carried out using a Bruker ELEXSYS E580 CW EPR spectrometer. The spectral simulations were performed using MATLAB 9.12.0.1884302 (R2022a) and the EasySpin 5.2.33 toolbox[53]. Cyclic voltammetry experiments were performed using a Gamry Instruments Reference 600 potentiostat. A standard three-electrode cell configuration was employed using a platinum disk working electrode, a platinum wire counter electrode, and a silver wire, separated by a Vycor tip, serving as the reference electrode. Formal redox potentials are referenced to the ferrocene/ferrocenium ($[Cp_2Fe]^{+/0}$) redox couple by using decamethylferrocene ($[Cp^*_2Fe]$; $E_{1/2} = -0.427$ V in THF) as an internal standard. Tetra($n$-butyl) ammonium hexafluorophosphate ($[nBu_4N][PF_6]$) was employed as the

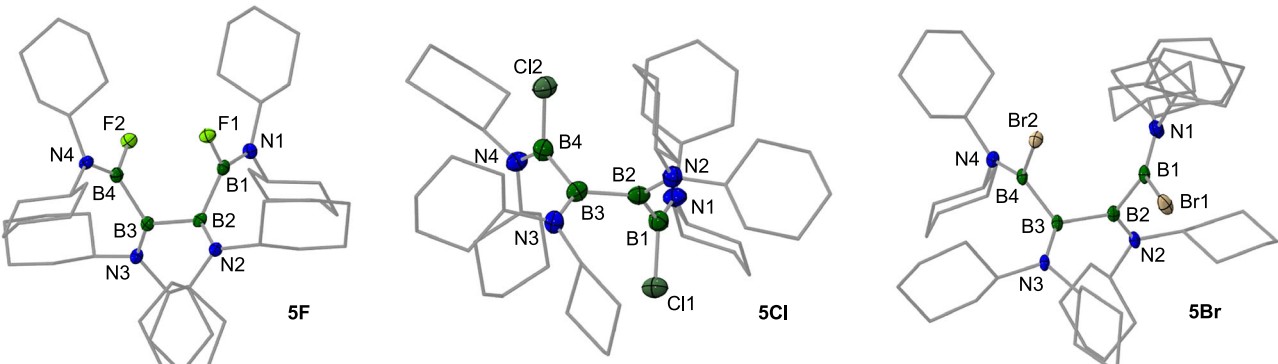

**Fig. 8 | Structures of linear ring-opening products 5F, 5Cl, and 5Br in the solid state.** Molecular structures of the linear tetraboranes **5 F**, **5Cl**, and **5Br** with displacement ellipsoids at 50% probability level (hydrogen atoms and ellipsoids of the cyclohexyl groups omitted for clarity).

**Fig. 9 | Overbromination of 1.** Synthesis of diborane $B_2(NCy_2)_2Br_2$ (**6**) by ring-opening reaction of **1** with excess $Br_2$.

supporting electrolyte. Compensation for resistive losses (iR drop) was employed for all measurements. Photolysis reactions were carried out on an Hg/Xe arc lamp ($I = 19\,A$, $U = 26\,V$, $P = 500\,W$, $\lambda = 210–600\,nm$) from the company LOT-QuantumDesign. Solvents, hexamethyldisilazane (HMDS), $Ph_2S_2$, $Ph_2Se_2$, $Ph_2Te_2$, $SbF_3$, $C_2Cl_6$, $Br_2$, and AgOTf (OTf = trifluoromethanesulfonate) were purchased from Sigma-Aldrich, ABCR or Alfa Aesar. [Ag][Al{OC(CF$_3$)$_3$}$_4$] was provided by the group of Prof. Ingo Krossing (University of Freiburg, Germany).

### Single-crystal X-ray diffraction

The crystallographic data of **2S**, **2Se**, **2Te**, [Cy$_2$NBO]$_3$, [**2S**][Al(OC(CF$_3$)$_3$)$_4$], **4S**, **5F**, **5Cl**, **5Br**, and **6** were collected on a XtaLAB Synergy Dualflex HyPix diffractometer with a Hybrid Pixel array detector and multi-layer mirror monochromated Cu$_{K\alpha}$ or Mo$_{K\alpha}$ radiation. The structures were solved using the intrinsic phasing method[54], refined with the ShelXL program[55] and expanded using Fourier techniques. All non-hydrogen atoms were refined anisotropically. Hydrogen atoms were included in structure factor calculations. All hydrogen atoms were assigned to idealized geometric positions.

### Computational details

All computations were performed using the Gaussian16 (Revision C.01) package[47]. All structures were fully optimized without symmetry constraints at the $\omega$B97xd level of theory employing def2-SVPP basis sets for all atoms[56,57]. Zero-point vibrational energies and thermal corrections were computed from frequency calculations with a standard state of 298 K and 1 atm; thermal free energies ($\Delta E_{298}$) were obtained from these single-point frequency calculations. The presence of true energy minima on the potential energy surface was verified for all optimized species by the absence of imaginary frequencies. For EPR, NMR, and orbital calculations def2-TZVP basis sets were used, combined with the SMD solvation model (scrf=smd) for inclusion of tetrahydrofuran solvent effects[58]. Calculated $^{19}F$ NMR chemical shifts were referenced to CCl$_3$F, the commonly used external standard in NMR spectroscopy, by first referencing to the calculated chemical shift of optimized $C_6F_6$ and putting the values into relation to its experimentally observed value of $\delta = -164.9$ ppm. Illustrations of optimized structures, as well as orbital and spin density plots were prepared with IQmol 3.1.3[59].

### Synthesis of $B_4(NCy_2)_4$ (1)

$B_4(NCy_2)_4$ (**1**) was prepared by the following procedure[23]: (Cy$_2$N)BCl$_2$ (200 mg, 760 µmol) and sodium sand (69.9 mg, 3.04 mmol, 4.0 equiv.) were combined in pentane (10 mL) and stirred for 3 d at room temperature, during which time a purple color gradually developed, accompanied by the precipitation of a purple solid. After filtration, all volatile components were removed in vacuo, and the resulting residue was washed with DME ($3 \times 5$ mL) and dried under reduced pressure. Thus, **1** was isolated as a blue solid (60.0 mg, 78.4 µmol, 41%). $^1H\{^{11}B\}$ NMR (400.1 MHz, C$_6$D$_6$, 297 K): $\delta = 3.39–3.27$ (m, 8H, C$H$), 2.04–1.94 (m, 16H, o-C$H_2$), 1.88–1.80 (m, 16H, m-C$H_2$), 1.71–1.58 (m, 24H, o-/p-C$H_2$), 1.46–1.32 (m, 16H, m-C$H_2$), 1.25–1.12 (m, 8H, p-C$H_2$) ppm. $^{13}C\{^1H\}$ NMR (100.7 MHz, C$_6$D$_6$, 297 K): $\delta = 63.9$ (C$H$), 35.7 (o-C$H_2$), 26.9 (m-C$H_2$), 26.2 (p-C$H_2$) ppm. $^{11}B$ NMR (128.5 MHz, C$_6$D$_6$, 297 K): $\delta = 68.0$ (br s) ppm. HRMS LIFDI for [C$_{48}$H$_{88}$N$_4$B$_4$]$^+$ = [M]$^+$: calc. 764.7376, found 764.7372.

### Synthesis of $B_4(NCy_2)_4S$ (2S)

$B_4(NCy_2)_4$ (**1**, 100 mg, 131 µmol) and diphenyl disulfide (14.3 mg, 65.4 µmol, 0.5 equiv.) were combined in benzene (6 mL) and stirred for 2 d under UV irradiation, resulting in the decolorization of the blue solution. After removing all volatiles from the reaction mixture under vacuum, the residue was washed with DME ($4 \times 5$ mL), filtered, and then dried under reduced pressure. This yielded product **2S** as a colorless solid (85.0 mg, 106 µmol, 81%). Single crystals suitable for X-ray diffraction analysis were obtained by recrystallization from hexane at $-30\,°C$. Note: When **1** is reacted with diphenyl sulfide, **2S** is obtained in 74% yield. $^1H\{^{11}B\}$ NMR (600.1 MHz, C$_6$D$_6$, 297 K): $\delta = 3.39–3.30$ (m, 4H, C$H$), 3.26–3.14 (m, 2H, C$H_2$), 3.07–3.00 (m, 2H, C$H$), 2.97–2.76 (m, 2H, C$H$), 2.02–1.63 (m, 40H, C$H_2$), 1.61–1.47 (m, 12H, C$H_2$), 1.44–1.18 (m, 22H, C$H_2$), 1.13–1.00 (m, 4H, C$H_2$) ppm. $^{13}C\{^1H\}$ NMR (125.7 MHz, C$_6$D$_6$, 297 K): $\delta = 66.3$ (C$H$), 66.1 (C$H$), 60.5 (C$H$), 57.0 (C$H$), 36.5 (C$H_2$), 36.0 (C$H_2$), 34.8 (C$H_2$), 34.0 (C$H_2$), 33.8 (C$H_2$), 33.5 (C$H_2$), 33.3 (C$H_2$), 32.7 (C$H_2$), 27.6 (C$H_2$), 27.5 (C$H_2$), 27.2 (C$H_2$), 27.1 (C$H_2$), 27.1 (C$H_2$), 26.6 (C$H_2$), 26.5 (C$H_2$), 26.3 (C$H_2$), 26.3 (C$H_2$), 26.2 (C$H_2$), 26.0 (C$H_2$) ppm. $^{11}B$ NMR (160.5 MHz, C$_6$D$_6$, 297 K): $\delta = 55.5$ (br s), 45.9 (br s) ppm. HRMS LIFDI for [C$_{48}$H$_{88}$B$_4$N$_4$S]$^+$ = [M]$^+$: calc. 796.7092; found 796.7096.

### Synthesis of $B_4(NCy_2)_4Se$ (2Se)

$B_4(NCy_2)_4$ (**1**, 100 mg, 131 µmol) and diphenyl diselenide (20.4 mg, 65.4 µmol, 0.5 equiv.) were combined in benzene (5 mL) and stirred for 2 d at 80 °C, resulting in a color change of the reaction mixture from blue to yellow. After removing all volatiles from the reaction mixture under vacuum, the residue was washed with DME ($4 \times 5$ mL), filtered, and dried under reduced pressure. This yielded product **2Se** as a colorless solid (66.3 mg, 78.6 µmol, 60%). Single crystals suitable for X-ray diffraction analysis were obtained by recrystallization from hexane at $-30\,°C$. $^1H\{^{11}B\}$ NMR (600.1 MHz, C$_6$D$_6$, 297 K): $\delta = 3.40–3.32$ (m, 4H, C$H$), 3.08–3.00 (m, 2H, C$H$), 2.99–2.91 (m, 1H, C$H$), 2.88–2.78 (m, 2H, C$H$), 2.10–2.04 (m, 4H, C$H_2$), 1.99–1.63 (m, 38H, C$H_2$), 1.61–1.46 (m, 12H, C$H_2$), 1.43–1.17 (m, 22H, C$H_2$), 1.13–0.98 (m, 4H, C$H_2$) ppm. $^{13}C\{^1H\}$ NMR (150.9 MHz, C$_6$D$_6$, 297 K): $\delta = 67.3$ (C$H$), 66.6 (C$H$), 60.8 (C$H$), 57.5 (C$H$), 36.7 (C$H_2$), 36.3 (C$H_2$), 35.0 (C$H_2$), 33.8 (C$H_2$), 33.2 (C$H_2$), 32.6 (C$H_2$), 27.5 (C$H_2$), 27.4 (C$H_2$), 27.3 (C$H_2$), 27.2 (C$H_2$), 27.2 (C$H_2$), 27.1 (C$H_2$), 26.6 (C$H_2$), 26.5 (C$H_2$), 26.3 (C$H_2$), 26.3 (C$H_2$), 26.2 (C$H_2$), 25.9 (C$H_2$) ppm. $^{11}B$ NMR (192.6 MHz, C$_6$D$_6$, 297 K): $\delta = 55.4$ (br s), 46.8 (br s) ppm. $^{77}Se\{^{11}B,^1H\}$ NMR (114.5 MHz, C$_6$D$_6$, 297 K): $\delta = 254$ ppm. HRMS LIFDI for [C$_{48}$H$_{88}$B$_4$N$_4$Se]$^+$ = [M]$^+$: calc. 843.6569; found 843.6577.

### Synthesis of $B_4(NCy_2)_4Te$ (2Te)

$B_4(NCy_2)_4$ (**1**, 60.0 mg, 78.5 µmol) and diphenyl ditelluride (16.1 mg, 39.2 µmol, 0.5 equiv.) were combined in benzene (5 mL) and stirred for 1 d under UV irradiation, resulting in a color change of the reaction mixture from blue to brown. After removal of all volatiles from the reaction mixture in vacuo, the residue was washed with DME ($3 \times 5$ mL), filtered, and dried under reduced pressure. This yielded product **2Te** as a colorless solid (34.6 mg, 38.8 µmol, 49%). Single crystals suitable for X-ray diffraction analysis were obtained by recrystallization from THF at $-30\,°C$. $^1H\{^{11}B\}$ NMR (600.1 MHz, C$_6$D$_6$, 297 K): $\delta = 3.56–3.35$ (m, 6H, C$H$ and C$H_2$), 3.11–2.95 (m, 4H, C$H$), 2.86–2.75 (m, 2H, C$H_2$), 2.28–2.20 (m, 2H, C$H_2$), 2.01–1.46 (m, 52H, C$H_2$), 1.40–1.15 (m, 18H, C$H_2$), 1.12–0.98 (m, 4H, C$H_2$) ppm. $^{13}C\{^1H\}$ NMR (150.9 MHz, C$_6$D$_6$, 297 K): $\delta = 68.7$ (C$H$), 66.9 (C$H$), 61.1 (C$H$), 57.6 (C$H$), 36.5 (C$H_2$), 36.5 (C$H_2$), 35.0 (C$H_2$), 33.6 (C$H_2$), 33.2 (C$H_2$), 32.8 (C$H_2$), 32.7 (C$H_2$), 32.6 (C$H_2$), 27.1 (C$H_2$), 27.0 (C$H_2$), 27.0 (C$H_2$), 26.9 (C$H_2$), 26.2 (C$H_2$), 26.0 (C$H_2$), 25.9 (C$H_2$) ppm. $^{11}B$ NMR (192.6 MHz, C$_6$D$_6$, 297 K): $\delta = 58.7$ (br s), 44.5 (br s) ppm. $^{125}Te$ NMR (189.3 MHz, C$_6$D$_6$, 297 K): $\delta = 15.6$ ppm. HRMS LIFDI for [C$_{48}$H$_{88}$B$_4$N$_4$Te]$^+$ = [M]$^+$: calc. 893.6443; found 893.6481.

### Synthesis of $B_2(NCy_2)_2S_2$ (3S)

$B_4(NCy_2)_4$ (**1**, 30.0 mg, 39.2 µmol) and S$_8$ (5.03 mg, 19.6 µmol, 0.5 equiv.) were combined in benzene (3 mL) and stirred for 5 d at 85 °C,

resulting in a color change of the reaction mixture from blue to yellow. After removing all volatiles from the reaction mixture under vacuum, the residue was washed with pentane (2 × 2 mL), filtered, and dried under reduced pressure. This yielded **3S** as a colorless solid (5.30 mg, 11.9 µmol, 15%). Single crystals suitable for X-ray diffraction analysis were obtained by recrystallization from benzene at room temperature. $^1H\{^{11}B\}$ NMR (400.1 MHz, $C_6D_6$, 297 K): $\delta = 3.39-3.23$ (m, 4H, C$H$), 1.95–1.70 (m, 16H, $o$-C$H_2$), 1.68–1.59 (m, 8H, m-C$H$ or $p$-CH$_2$), 1.51–1.40 (m, 4H, m-C$H$ or $p$-CH$_2$), 1.24–1.10 (m, 8H, $m$-C$H$ or $p$-CH$_2$), 1.05–0.91 (m, 4H, $m$-C$H$ or $p$-CH$_2$ ppm. $^{13}C\{^1H\}$ NMR (100.7 MHz, $C_6D_6$, 297 K): $\delta = 58.2$ (CH), 34.0 ($o$-CH$_2$), 26.7 (CH$_2$), 25.8 (CH$_2$) ppm. $^{11}B$ NMR (128.5 MHz, $C_6D_6$, 297 K): $\delta = 37.5$ (br s) ppm. HRMS LIFDI for $[C_{24}H_{44}B_2N_2S_2]^+ = [M]^+$: calc. 446.3127; found 446.3119.

## Synthesis of $[B_4(NCy_2)_4S][Al\{OC(CF_3)_3\}_4]$ ($[2S][Al\{OC(CF_3)_3\}_4]$)

To a solution of **2S** (100 mg, 125 µmol) in 1,2-difluorobenzene (8 mL), $[Ag][Al(OC(CF_3)_3)_4]$ (134 mg, 125 µmol, 1.0 equiv.) was added and stirred for 20 min at room temperature under exclusion of light, resulting in the reaction mixture turning dark green. The insoluble materials in the reaction mixture were subsequently removed by filtration, and the solvent from the green filtrate was evaporated under reduced pressure. The residue was washed with benzene (2 × 5 mL) and dried under reduced pressure, yielding $[2S][Al\{OC(CF_3)_3\}_4]$ as a dark green solid (175 mg, 99.2 µmol, 79%). Single crystals suitable for X-ray diffraction analysis were obtained by crystallization in a mixture of toluene and DME at −30 °C. HRMS LIFDI for $[C_{48}H_{88}B_4N_4S]^+ = [M]^+$: calc. 796.7096; found 796.7089. UV-vis (1,2-difluorobenzene): $\lambda_1 = 422$ nm, $\lambda_2 = 673$ nm (shoulder).

## Synthesis of $B_4(NCy_2)_4S(OTf)_2$ (4S)

$B_4(NCy_2)_4S$ (**2S**, 50 mg, 62.8 µmol) and AgOTf (32.3 mg, 125.5 µmol, 2.0 equiv.) were combined in benzene (4 mL) and stirred for 17 h at 50 °C, causing the reaction mixture to change from colorless to yellow. The insoluble materials in the reaction mixture were separated by filtration. After the removal of all volatiles from the reaction mixture in vacuo, the residue was extracted with cold pentane (4 × 1 mL, −30 °C), filtered again, and the solvent from the filtrate was removed under reduced pressure. The extraction in cold pentane was repeated until the filtrate became colorless, yielding **4S** as a colorless solid (23.0 mg, 21.0 µmol, 33%). Single crystals suitable for X-ray diffraction analysis were obtained by crystallization in toluene at −30 °C. Note: In solution, a mixture of temperature-dependent conformational isomers is present, which cannot be distinguished from each other. $^1H\{^{11}B\}$ NMR (600.1 MHz, $d_8$-toluene, 298.15 K): $\delta = 4.17-4.02$ (m, 1H, C$H$), 3.52–3.39 (m, 2H, C$H$), 3.23–3.07 (m, 2H, C$H$), 2.82–2.60 (m, 2H, C$H$), 2.45–2.33 (m, 1H, C$H$), 2.06–0.82 (m, 80H, C$H_2$) ppm. $^{13}C\{^1H\}$ NMR (150.9 MHz, $d_8$-toluene, 293.15 K): $\delta = 131.4$ (CF$_3$), 122.4 (CF$_3$), 120.3 (CF$_3$), 118.2 (CF$_3$), 116.1 (CF$_3$), 71.1 (CH), 68.4 (CH), 66.3 (CH), 61.0 (CH), 60.8 (CH), 60.4 (CH), 59.8 (CH), 59.4 (CH), 57.0 (CH), 56.0 (CH), 56.0 (CH), 55.8 (CH), 55.2 (CH), 54.3 (CH), 54.1 (CH), 38.2 (CH$_2$), 37.8 (CH$_2$), 37.1 (CH$_2$), 36.6 (CH$_2$), 36.0 (CH$_2$), 35.1 (CH$_2$), 34.7 (CH$_2$), 34.5 (CH$_2$), 34.1 (CH$_2$), 33.9 (CH$_2$), 33.8 (CH$_2$), 33.8 (CH$_2$), 33.7 (CH$_2$), 33.6 (CH$_2$), 33.5 (CH$_2$), 33.2 (CH$_2$), 32.9 (CH$_2$), 32.5 (CH$_2$), 32.1 (CH$_2$), 31.4 (CH$_2$), 29.1 (CH$_2$), 28.0 (CH$_2$), 27.7 (CH$_2$), 27.6 (CH$_2$), 27.6 (CH$_2$), 27.4 (CH$_2$), 27.3 (CH$_2$), 27.2 (CH$_2$), 27.2 (CH$_2$), 27.2 (CH$_2$), 27.1 (CH$_2$), 27.1 (CH$_2$), 27.1 (CH$_2$), 27.0 (CH$_2$), 26.9 (CH$_2$), 26.7 (CH$_2$), 26.6 (CH$_2$), 26.6 (CH$_2$), 26.6 (CH$_2$), 26.5 (CH$_2$), 26.5 (CH$_2$), 26.4 (CH$_2$), 26.3 (CH$_2$), 26.3 (CH$_2$), 26.3 (CH$_2$), 26.2 (CH$_2$), 26.2 (CH$_2$), 26.1 (CH$_2$), 26.0 (CH$_2$), 25.7 (CH$_2$), 25.6 (CH$_2$), 25.4 (CH$_2$), 25.3 (CH$_2$), 25.1 (CH$_2$), 25.0 (CH$_2$), 24.8 (CH$_2$), 24.6 (CH$_2$) ppm. $^{11}B$ NMR (192.6 MHz, $d_8$-toluene, 293.15 K): $\delta = 42.1$ (br s), 34.4 (br s) ppm. $^{11}B$ NMR (192.6 MHz, $d_8$-toluene, 193.15 K): $\delta = 36.3$ (br s), 26.2 (br s) ppm. $^{19}F$ NMR (564.7 MHz, $d_8$-toluene, 293.15 K): $\delta = -74.99$, −76.29, −76.61, −76.93, −77.08, −78.20 ppm. $^{19}F$ NMR (564.7 MHz, $d_8$-toluene, 193.15 K): $\delta = -75.41$, −75.98, −76.16, −76.37, −76.48, −76.66, −76.76,

−76.83, −77.17, −77.39, −78.01 ppm. HRMS LIFDI for $[C_{49}H_{88}B_4F_3N_4O_3S_2]^+ = [M-CSF_3O_3]^+$: calc. 945.6617; found 945.6616.

## Synthesis of $B_4(NCy_2)_4F_2$ (5 F)

$B_4(NCy_2)_4$ (**1**, 50.0 mg, 65.4 µmol) and SbF$_3$ (23.4 mg, 131 µmol, 2.0 equiv.) were combined in a flask (silanized with HMDS) and dissolved in benzene (5 mL). After stirring for 20 h at 85 °C, the initially blue solution underwent decolorization. The insoluble materials in the reaction mixture were subsequently removed by filtration. After removal of all volatiles from the reaction mixture in vacuo, the residue was washed with DME (4 × 2 mL), filtered, and dried under reduced pressure. This yielded product **5 F** as a colorless solid (25.0 mg, 31.2 µmol, 47%). Single crystals suitable for X-ray diffraction analysis were obtained by crystallization in benzene at room temperature. Note: In solution, a mixture of temperature-dependent conformational isomers is present, which cannot be distinguished from each other. $^1H\{^{11}B\}$ NMR (600.1 MHz, $d_8$-toluene, 383.15 K): $\delta = 3.41-3.33$ (m, 2H, C$H$), 3.30–3.19 (m, 2H, C$H$), 3.16–3.08 (m, 2H, C$H$), 2.72–2.63 (m, 2H, C$H_2$), 2.20–2.09 (m, 2H, C$H_2$), 2.01–0.99 (m, 76H, C$H_2$) ppm. $^{13}C\{^1H\}$ NMR (150.9 MHz, $d_8$-toluene, 383.15 K): $\delta = 68.1$ (CH), 62.9 (CH), 58.6 (CH), 58.6 (CH), 55.5 (CH), 37.5 (CH$_2$), 35.4 (CH$_2$), 33.7 (CH$_2$), 28.0 (CH$_2$), 27.7 (CH$_2$), 27.5 (CH$_2$), 27.1 (CH$_2$), 26.8 (CH$_2$), 26.7 (CH$_2$), 26.6 (CH$_2$) ppm. $^{11}B$ NMR (192.6 MHz, $d_8$-toluene, 383.15 K): $\delta = 56.9$ (br s), 37.9 (br s) ppm. $^{19}F$ NMR (564.7 MHz, $d_8$-toluene, 403.15 K): $\delta = -79.81$ ppm. $^{19}F$ NMR (564.7 MHz, $d_8$-toluene, 298.15 K): $\delta = -76.18$, −77.04, −84.96, −87.05, −88.53 ppm. HRMS LIFDI for $[C_{48}H_{88}B_4F_2N_4]^+ = [M]^+$: calc. 802.7344; found 802.7329.

## Synthesis of $B_4(NCy_2)_4Cl_2$ (5Cl)

$B_4(NCy_2)_4$ (**1**, 50 mg, 65.4 µmol) and hexachloroethane (15.5 mg, 65.4 µmol, 1.0 equiv.) were combined in benzene (3 mL) and stirred for 3 h at 60 °C, resulting in the decolorization of the blue solution. The insoluble materials in the reaction mixture were subsequently removed by filtration, and the solvent of the filtrate was removed under reduced pressure. This produced **5Cl** as a colorless solid (43.0 mg, 51.5 µmol, 78%). Single crystals suitable for X-ray diffraction analysis were obtained by crystallization in DME at −30 °C. Note: Selective oxidation of **1** to **5Cl** was also observed when **1** was reacted with one equivalent of AlCl$_3$ or GaCl$_3$. $^1H\{^{11}B\}$ NMR (600.1 MHz, $C_6D_6$, 297 K): $\delta = 3.97-3.88$ (m, 2H, C$H$), 3.79–3.67 (m, 2H, C$H$), 3.60–3.52 (m, 2H, C$H$), 3.06–2.54 (m, 4H, C$H$ and C$H_2$), 2.43–0.93 (m, 78H, C$H_2$) ppm. $^{13}C\{^1H\}$ NMR (150.9 MHz, $d_8$-toluene, 383.15 K): $\delta = 68.0$ (CH), 67.1 (CH), 63.7 (CH), 57.8 (CH), 39.2 (CH$_2$), 38.7 (CH$_2$), 35.8 (CH$_2$), 35.4 (CH$_2$), 34.5 (CH$_2$), 34.1 (CH$_2$), 33.6 (CH$_2$), 28.0 (CH$_2$), 28.0 (CH$_2$), 27.9 (CH$_2$), 27.9 (CH$_2$), 27.8 (CH$_2$), 27.2 (CH$_2$), 27.1 (CH$_2$), 27.0 (CH$_2$), 26.7 (CH$_2$), 26.5 (CH$_2$), 26.4 (CH$_2$) ppm. $^{11}B$ NMR (192.6 MHz, $d_8$-toluene, 383.15 K): $\delta = 57.5$ (br s), 43.4 (br s) ppm. HRMS LIFDI for $[C_{48}H_{88}B_4Cl_2N_4]^+ = [M]^+$: calc. 834.6751; found 834.6753.

## Synthesis of $B_4(NCy_2)_4Br_2$ (5Br)

To a solution of $B_4(NCy_2)_4$ (**1**, 50 mg, 65.4 µmol) in pentane (20 mL) cooled to −60 °C, a pentane solution (5 mL) of Br$_2$ (10.5 mg, 0.20 mL of 0.325 M in benzene 65.4 µmol, 1.0 equiv.) at −60 °C was added dropwise. The reaction mixture was stirred for 24 h while slowly warming to room temperature, resulting in the decolorization of the blue solution. After removal of all volatiles from the reaction mixture in vacuo, the residue was washed with cold hexane (2 mL, −60 °C) and dried under reduced pressure. This yielded **5Br** as a colorless solid (43.0 mg, 46.5 µmol, 71%). Single crystals for X-ray diffraction analysis were obtained by crystallization in pentane at −30 °C. $^1H\{^{11}B\}$ NMR (600.1 MHz, $d_8$-toluene, 383.15 K): $\delta = 3.94-3.88$ (m, 2H, C$H$), 3.65–3.56 (m, 4H, C$H$), 3.51–3.27 (m, 2H, C$H$), 3.63–2.44 (m, 2H, C$H_2$), 3.43–2.35 (m, 2H, C$H_2$), 2.28–2.20 (m, 2H, C$H_2$), 2.15–2.09 (m, 2H, C$H_2$), 1.94–0.99

(m, 72H, C$H_2$) ppm. $^{13}$C{$^1$H} NMR (150.9 MHz, $d_8$-toluene, 383.15 K): $\delta$ = 68.6 (CH), 66.8 (CH), 64.8 (CH), 59.3 (CH), 39.0 (C$H_2$), 38.9 (C$H_2$), 35.9 (C$H_2$), 35.4 (C$H_2$), 34.5 (C$H_2$), 34.1 (C$H_2$), 33.8 (C$H_2$), 28.0 (C$H_2$), 27.9 (C$H_2$), 27.8 (C$H_2$), 27.7(C$H_2$), 27.1 (C$H_2$), 27.0 (C$H_2$), 26.9 (C$H_2$), 26.7 (C$H_2$), 26.5 (C$H_2$), 26.4 (C$H_2$) ppm. $^{11}$B NMR (192.6 MHz, $d_8$-toluene, 383.15 K): $\delta$ = 57.0 (br s), 43.3 (br s) ppm. HRMS LIFDI for [C$_{48}$H$_{88}$B$_4$BrN$_4$]$^+$ = [M−Br]$^+$: calc. 844.6575; found 844.6564.

### Synthesis of B$_4$(NCy$_2$)$_4$(OTf)$_2$ (5OTf)

B$_4$(NCy$_2$)$_4$ (**1**, 50 mg, 65.4 μmol) and AgOTf (33.6 mg, 130.8 μmol, 2.0 equiv.) were combined in benzene (4 mL) and stirred for 1 h at room temperature, resulting in the decolorization of the blue solution. The insoluble materials in the reaction mixture were subsequently separated by filtration, and the solvent of the filtrate was removed under reduced pressure, yielding **5OTf** as a colorless solid (64.0 mg, 65.4 μmol, 92%). Note: Single crystals of **5OTf** suitable for X-ray diffraction analysis could not be obtained despite numerous crystallization attempts. Attempts involved systematic variations of solvents, temperatures, and crystallization techniques, yet the resulting crystals consistently exhibited excessive disorder, preventing reliable X-ray diffraction analysis. Note: In solution, a mixture of temperature-dependent conformational isomers is present, which cannot be distinguished from one another. $^1$H{$^{11}$B} NMR (600.1 MHz, C$_6$D$_6$, 393.15 K): $\delta$ = 4.01–3.94 (m, 1H, CH), 3.76–3.55 (m, 4H, CH), 3.29–3.22 (m, 1H, CH), 3.16–3.05 (m, 1H, CH), 3.00–2.92 (m, 1H, CH), 2.54–2.43 (m, 3H, C$H_2$), 2.37–2.31 (m, 1H, C$H_2$), 2.25–2.16 (m, 4H, C$H_2$), 2.06–1.02 (m, 72H, C$H_2$) ppm. $^{11}$B NMR (192.6 MHz, $d_8$-toluene, 393.15 K): $\delta$ = 55.5 (br), 36.2 (br) ppm. $^{13}$C{$^1$H,$^{11}$B} NMR (150.9 MHz, $d_8$-toluene, 383.15 K): $\delta$ = 120.7 (CF$_3$), 118.6 (CF$_3$), 71.8 (CH), 67.6 (CH), 67.5 (CH), 60.1 (CH), 60.1 (CH), 57.4 (CH), 56.8 (CH), 39.0 (C$H_2$), 38.7 (C$H_2$), 38.1 (C$H_2$), 37.5 (C$H_2$), 36.3 (C$H_2$), 36.1 (C$H_2$), 35.8 (C$H_2$), 34.2 (C$H_2$), 33.6 (C$H_2$), 31.2 (C$H_2$), 30.2 (C$H_2$), 28.5 (C$H_2$), 28.4 (C$H_2$), 28.2 (C$H_2$), 27.9 (C$H_2$), 27.8 (C$H_2$), 27.6 (C$H_2$), 27.2 (C$H_2$), 27.0 (C$H_2$), 26.9 (C$H_2$), 26.7 (C$H_2$), 26.3 (C$H_2$), 26.2 (C$H_2$), 26.0 (C$H_2$), 25.9 (C$H_2$), 25.9 (C$H_2$) ppm. $^{19}$F NMR (470.6 MHz, $d_8$-toluene, 297.15 K): $\delta$ = −75.66, −75.83, −75.95, −76.20, −77.26, −78.10 ppm. $^{19}$F NMR (564.7 MHz, $d_8$-toluene, 173.15 K): $\delta$ = −76.20, −77.62, −77.81 ppm. HRMS LIFDI for [C$_{49}$H$_{88}$B$_4$N$_4$F$_3$O$_3$S]$^+$ = [M−CSF$_3$O$_3$]$^+$: calc. 913.6896; found 913.6889.

### Synthesis of B$_2$(NCy$_2$)$_2$Br$_2$ (6)

To a solution of B$_4$(NCy$_2$)$_4$ (**1**, 50 mg, 65.4 μmol) in benzene (10 mL), a chilled solution of Br$_2$ (20.9 mg, 0.40 mL of 0.235 M in benzene, 130 μmol, 2.0 equiv.) in benzene at 0 °C was added and stirred for 2 h. During this time, decolorization of the blue reaction mixture was observed. After removal of all volatiles from the reaction mixture in vacuo, the residue was washed with cold hexane (2 mL, −30 °C) and dried under reduced pressure, yielding **6** as a colorless solid (39.8 mg, 73.4 μmol, 56%). Single crystals suitable for X-ray diffraction analysis were obtained by crystallization in hexane at −30 °C. $^1$H{$^{11}$B} NMR (600.1 MHz, $d_8$-toluene, 383.15 K): $\delta$ = 3.36–3.30 (m, 2H, CH), 3.28–3.21 (m, 2H, CH), 2.44–2.33 (m, 4H, C$H_2$), 1.99–1.93 (m, 2H, C$H_2$), 1.72–1.60 (m, 16H, C$H_2$), 1.52–1.41 (m, 6H, C$H_2$), 1.29–1.06 (m, 10H, C$H_2$), 1.02–0.91 (m, 2H, C$H_2$) ppm. $^{13}$C{$^1$H} NMR (150.9 MHz, $d_8$-toluene, 383.15 K): $\delta$ = 65.5 (CH), 58.8 (CH), 33.6 (C$H_2$), 33.5 (C$H_2$), 27.1 (C$H_2$), 27.0 (C$H_2$), 26.7 (C$H_2$), 26.5 (C$H_2$), 25.9 (C$H_2$), 25.8 (C$H_2$) ppm. $^{11}$B NMR (192.6 MHz, $d_8$-toluene, 383.15 K): $\delta$ = 37.8 (s) ppm. HRMS LIFDI for [C$_{24}$H$_{44}$B$_2$Br$_2$N$_2$]$^+$ = [M]$^+$: calc. 542.2031; found 542.2018.

### Data availability

All data are available from the corresponding author upon request. Experimental details, procedures, spectra, crystallographic, and computational details are provided in the Supplementary Information file. Source Data (cartesian coordinates for calculated structures) are provided with this manuscript. CCDC 2423870-2423879 contain the supplementary crystallographic data for this paper. These data can be obtained free of charge via www.ccdc.cam.ac.uk/data_request/cif, or by emailing data_request@ccdc.cam.ac.uk, or by contacting The Cambridge Crystallographic Data Centre, 12 Union Road, Cambridge CB2 1EZ, UK; fax: +44 1223 336033. Source data are provided with this paper.

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

## Acknowledgements

The Julius-Maximilians-Universität of Würzburg and the Deutsche Forschungsgemeinschaft (grant numbers BR1149/21-1 and 466754611) are acknowledged for financially supporting this work. We are grateful for the Open Access funding enabled and organized by Projekt DEAL.

## Author contributions

Experiments were designed by E.B. and performed by E.B. with contributions from D.B. Data analysis was performed by E.B. and I.K. X-ray crystallography was carried out by M. D., T.W., and C.M. Computational studies were performed by T.K. EPR and cyclic voltammetry studies were conducted by I.K. The manuscript was written and edited by I.K. and T.K. with input from all authors. H.B. supervised the investigation.

## Funding

## Competing interests

The authors declare no competing interests.
