## [Transparent Peer Review file · Nature Communications]

Boron-Chalcogen Heterocycles and Linear Tetraboranes from a Cyclic Tetra(amino)tetraborane

Corresponding Author: Professor Holger Braunschweig

Version 0:

Reviewer comments:

Reviewer #1

(Remarks to the Author)

In this article, Braunschweig et al. described the reactivity of a cyclic tetra(amino)tetraborane toward diphenyl dichalcogenide reagents, chalcogens, and halogenating agents. These transformations involve ring-expansion and ring-opening mechanisms, affording the formation of novel five-membered B4E heterocycles and linear tetraboranes. Despite this is another contribution after their most recent publication on the redox chemistry of tetra(amino)tetraborane (Chem. 2025, 11, 102338), this reviewer conclude that it is a very important advance in main group chemistry. Thus, I recommend a publication of this manuscript unless the following minor issues have been addressed.

1) the authors state this work presents the first investigation into the reactivity of cyclic tetraboranes is inappropriate.

2) Many typos should be corrected.

P1 line 34: research, P2 line 29 structurally ----

3) I wonder which product would be generated if oxidation of 2S with two equivalents of $[Ag][Al\{OC(CF_3)_3\}_4]$.

4) it seems unreasonable 2Se and 2Te are inert toward $AgOTf$ considering their similar oxidation potentials with respect to that for 2S, some comments should be included.

Reviewer #2

(Remarks to the Author)

This manuscript reports interesting ring opening and ring expansion reactions starting with the previously reported cyclic tetraboron compound tetrakis(dicyclohexylamino)tetraborane. Oxidative insertion of S, Se or Te into one of the B-B bonds leads to ring expansion to a five-membered ring. Oxidative ring opening leads to new tetraboron chain compounds that are functionalized at both ends and might be suitable starting points for further reactions. I congratulate the authors to these nice results. The work is carried out with great competence and the manuscript is written in an intelligible style. I think that with regard to the novelty and the impact it is a borderline case, but would support its publication since I think that the new compounds are excellent reagents for further exciting chemistry. I would recommend publication without any reservation if the authors could show how the products of ring opening or expansion could be further used. Here are some suggestions.

1) The five-membered ring with sulphur is surprisingly stable. What happens if it is reacted with Cu(I), for example $CuBF_4$? Is the sulphur atom in the ring sufficiently Lewis basic to bind to metals? Could these rings be used as ligands?

2) The ring opening leads to four-membered boron chains. Could the functionalization be used to obtain longer boron chains? What else is possible?

3) Is there a chance for a ring-opening polymerization reaction using suitable catalysts and temperature?

Maybe the authors could answer to one of these points. Then, I think that the manuscript would become suitable for Nat. Commun. without any reservations.

Reviewer #3

(Remarks to the Author)

This paper reports the chemistry of a cyclic B₄ compound. Because of its electron deficiency, boron does not usually form cyclic compounds like carbon. The cyclic tetra(amino)tetraborane is a rare cyclic boron molecule, in which the electron deficiency of boron is compensated by the extra electron in the N atom of the amino ligands, essentially, using the C₂ and BN isolobal analogy. In a recent report (ref 23), the authors studied the redox chemistry of the B₄ compound. In the present

paper, the authors explore the ring expansion reactions of the B₄ compounds using chalcogens (E = S, Se, Te). They observed both ring expansion to form EB₄ hetero cycles and ring opening reactions to form linear B-B-E-B-B compounds. This is an interesting investigation by one of the most preeminent lab in boron chemistry. The present paper reports interesting chemistry and strategies to expand the boron rings. It is suitable for Nat. Commun. I only have one comment for the authors to consider before acceptance.

Both the B₄ cycle in the parent tetra(amino)tetraborane and the hetero-EB₄ cycle are nonplanar. It would be interesting to discuss the bonding and explain clearly the driving force for nonplanarity, and point it out in the abstract. The connection with the corresponding hydrocarbon compounds using the C₂/BN isolobal analogy would be interesting to mention and discuss, in particular, in terms of the chemical bonding and structural differences.

Version 1:

Reviewer comments:

Reviewer #1

(Remarks to the Author)

The authors have addressed all my concerns and this manuscript is recommended for publication in current version.

Reviewer #2

(Remarks to the Author)

The authors have addressed all points. Although only minor additions were made to the manuscript, I support its acceptance without further changes.

Reviewer #3

(Remarks to the Author)

While I think some brief discussion of the structure and bonding would have improved the paper, I am not going to insist on that. I recommend acceptance.

POINT-BY-POINT RESPONSE TO THE REVIEWERS' COMMENTS

Reviewer #1:

In this article, Braunschweig et al. described the reactivity of a cyclic tetra(amino)tetraborane toward diphenyl dichalcogenide reagents, chalcogens, and halogenating agents. These transformations involve ring-expansion and ring-opening mechanisms, affording the formation of novel five-membered B₄E heterocycles and linear tetraboranes. Despite this is another contribution after their most recent publication on the redox chemistry of tetra(amino)tetraborane (Chem. 2025, 11, 102338), this reviewer conclude that it is a very important advance in main group chemistry. Thus, I recommend a publication of this manuscript unless the following minor issues have been addressed.

1) the authors state this work presents the first investigation into the reactivity of cyclic tetraboranes is inappropriate.

RESPONSE: We can see that the notion "first", as used in the abstract, is indeed somewhat misleading, considering the facts that (i) we have quite recently already outlined the redox chemistry of B₄(NCy₂)₄, and that (ii) some reactivity is already known for related yet nonclassical cyclic tetraboranes (which we consider highly different in nature). Thus, we have altered the text accordingly, avoiding this expression.

2) Many typos should be corrected.

P1 line 34: research, P2 line 29 strucutally ----

RESPONSE: We have double-checked the manuscript again and hope to have eliminated most typos.

3) I wonder which product would be generated if oxidation of 2S with two equivalents of [Ag][Al{OC(CF₃)₃}₄]₄.

RESPONSE: This is indeed a very good question, and we have tried this reaction repeatedly aiming at the generation of a dication of **2S**. Unfortunately, all efforts remained without success, and only the radical cation of **2S** was detected. We added a note to the manuscript. By contrast, subsequent reaction of the isolated radical cation with other powerful oxidizing reagents such as [NO][SbF₆] lacked any selectivity. Also, when switching from [Ag][Al{OC(CF₃)₃}₄]₄ to AgOTf, only ring-opening and formation of **4S** occurred.

4) it seems unreasonable 2Se and 2Te are inert toward AgOTf considering their similar oxidation potentials with respect to that for 2S, some comments should be included.

RESPONSE: **2Se** and **2Te** are indeed inert UNDER THE SAME CONDITIONS. However, **2Se** does react with AgOTf at higher temperatures (80°C). In addition, longer reaction times (6 d) are required, and the reaction is highly unselective. No tractable materials could be isolated. **2Te**, by contrast, does not react at all, even at 100°C. We have added these details to the manuscript.

Reviewer #2:

This manuscript reports interesting ring opening and ring expansion reactions starting with the previously reported cyclic tetraboron compound tetrakis(dicyclohexylamino)tetraborane. Oxidative insertion of S, Se or Te into one of the B-B bonds leads to ring expansion to a five-membered ring. Oxidative ring opening leads to new tetraboron chain compounds that are functionalized at both ends and might be suitable starting points for further reactions. I congratulate the authors to these nice results. The work is carried out with great competence and the manuscript is written in an intelligible style. I think that with regard to the novelty and the impact it is a borderline case, but would support its publication since I think that the new compounds are excellent reagents for further exciting chemistry. I would recommend publication without any reservation if the authors could show how the products of ring opening or expansion could be further used. Here are some suggestions.

1) The five-membered ring with sulphur is surprisingly stable. What happens if it is reacted with Cu(I), for example CuBF₄? Is the sulphur atom in the ring sufficiently Lewis basic to bind to metals? Could these rings be used as ligands?

RESPONSE: We thank the reviewer for the overall very positive feedback and these interesting thoughts. We have not put that much effort into this line of reactivity so far, even though we should probably do so in the future. Thus, we cannot provide an answer to the question "whether these rings can be used as ligands" yet. Also, we have not tested the reactivity of **2S** toward Cu(I) so far. But we have studied the reaction of **2S** toward AuCl. Unfortunately, the results are still inconclusive and require some more work before we feel comfortable to draw any conclusion. At 60°C, **2S** does react with Au(I), albeit without much selectivity. Several boron-containing species were observed by NMR, but nothing could be isolated yet.

2) The ring opening leads to four-membered boron chains. Could the functionalization be used to obtain longer boron chains? What else is possible?

RESPONSE: This is also an intriguing question. Actually, we have been searching for possibilities to efficiently generate longer boron chains for a long time. However, we believe that the ring-opening strategy presented in this paper is quite limited to a maximum of four boron atoms, due to the lack of suitable boron-based ring-opening reagents capable of adding boron atoms to those of the B₄ ring. Larger boron ring precursors on the other hand are even harder to prepare in the first place (only one B₆ is known) and would probably lack the required ring strain for any ring-opening reactivity. We have had no success in these directions so far. Therefore, we are currently focusing our efforts in this area of research to other approaches.

3) Is there a chance for a ring-opening polymerization reaction using suitable catalysts and temperature?

RESPONSE: This question has been in our mind as well, for quite a while now. However, we apologize for not being able to provide any answer yet, as we are still at the stage of identifying suitable ring-opening catalysts. But we are actively pursuing this line of research, hoping to gain first insights soon.

Maybe the authors could answer to one of these points. Then, I think that the manuscript would become suitable for Nat. Commun. without any reservations.

Reviewer #3:

This paper reports the chemistry of a cyclic B₄ compound. Because of its electron deficiency, boron does not usually form cyclic compounds like carbon. The cyclic tetra(amino)tetraborane is a rare cyclic boron molecule, in which the electron deficiency of boron is compensated by the extra electron in the N atom of the amino ligands, essentially, using the C₂ and BN isolobal analogy. In a recent report (ref 23), the authors studied the redox chemistry of the B₄ compound. In the present paper, the authors explore the ring expansion reactions of the B₄ compounds using chalcogens (E = S, Se, Te). They observed both ring expansion to form EB₄ hetero cycles and ring opening reactions to form linear B-B-E-B-B compounds. This is an interesting investigation by one of the most preeminent lab in boron chemistry. The present paper reports interesting chemistry and strategies to expand the boron rings. It is suitable for Nat. Commun. I only have one comment for the authors to consider before acceptance.

Both the B₄ cycle in the parent tetra(amino)tetraborane and the hetero-EB₄ cycle are nonplanar. It would be interesting to discuss the bonding and explain clearly the driving force for nonplanarity, and point it out in the abstract. The connection with the corresponding hydrocarbon compounds using the C₂/BN isolobal analogy would be interesting to mention and discuss, in particular, in terms of the chemical bonding and structural differences.

RESPONSE: We thank the reviewer for this positive feedback. Regarding the first part of the comment, we would like to point out that the nonplanar, butterfly-type structure of cyclic tetra(amino)tetraboranes has already been discussed in the literature including our previous work on the synthesis and redox chemistry of B₄(NCy₂)₄. Accordingly, the puckered structure is caused by strong B-B σ -bonding interactions, in part over more than two B atoms and/or spanning the amino N atoms. In addition, unpublished DFT results show that the butterfly-type configuration is dramatically lower in energy by $\Delta E = -55$ kcal/mol than a forced planar configuration. Overall, we believe that it is not necessary to include such a discussion in the present manuscript, and that it would not add much to it in general, as it is focused on reactivity studies. Regarding “The connection with the corresponding hydrocarbon compounds using the C₂/BN isolobal analogy”, we are not certain to what exactly the reviewer refers to, since the cyclotetraboranes do not contain N atoms to allow for such a comparison. If the reviewer would be willing to clarify this question, we can provide our thoughts on this.